# The Effect of Halloysite Nanotubes on the Fire Retardancy Properties of Partially Biobased Polyamide 610

**DOI:** 10.3390/polym12123050

**Published:** 2020-12-19

**Authors:** David Marset, Celia Dolza, Eduardo Fages, Eloi Gonga, Oscar Gutiérrez, Jaume Gomez-Caturla, Juan Ivorra-Martinez, Lourdes Sanchez-Nacher, Luis Quiles-Carrillo

**Affiliations:** 1Textile Industry Research Association (AITEX), Plaza Emilio Sala 1, 03801 Alcoy, Spain; dmarset@aitex.es (D.M.); cdolza@aitex.es (C.D.); EFages@aitex.es (E.F.); egonga@aitex.es (E.G.); ogutierrez@aitex.es (O.G.); 2Technological Institute of Materials (ITM), Universitat Politècnica de València (UPV), Plaza Ferrándiz y Carbonell 1, 03801 Alcoy, Spain; jaugoca@epsa.upv.es (J.G.-C.); lsanchez@mcm.upv.es (L.S.-N.)

**Keywords:** PA610, halloysite nanotubes (HNTs), nanocomposites, flame retardant, cone calorimeter

## Abstract

The main objective of the work reported here was the analysis and evaluation of halloysite nanotubes (HNTs) as natural flame retardancy filler in partially biobased polyamide 610 (PA610), with 63% of carbon from natural sources. HNTs are naturally occurring clays with a nanotube-like shape. PA610 compounds containing 10%, 20%, and 30% HNT were obtained in a twin-screw co-rotating extruder. The resulting blends were injection molded to create standard samples for fire testing. The incorporation of the HNTs in the PA610 matrix leads to a reduction both in the optical density and a significant reduction in the number of toxic gases emitted during combustion. This improvement in fire properties is relevant in applications where fire safety is required. With regard to calorimetric cone results, the incorporation of 30% HNTs achieved a significant reduction in terms of the peak values obtained of the heat released rate (HRR), changing from 743 kW/m^2^ to about 580 kW/m^2^ and directly modifying the shape of the characteristic curve. This improvement in the heat released has produced a delay in the mass transfer of the volatile decomposition products, which are entrapped inside the HNTs’ lumen, making it difficult for the sample to burn. However, in relation to the ignition time of the samples (TTI), the incorporation of HNTs reduces the ignition start time about 20 s. The results indicate that it is possible to obtain polymer formulations with a high renewable content such as PA610, and a natural occurring inorganic filler in the form of a nanotube, i.e., HNTs, with good flame retardancy properties in terms of toxicity, optical density and UL94 test.

## 1. Introduction

The current social awareness about the environmental problems derived from the use of non-renewable polymers and additives is generating a great change in the industry. Moreover, governments are beginning to become concerned about this problem and are starting to develop legislation that favors environmental protection and the use of materials that reduce the harmful impact on nature [1,2]. In particular, a great effort has been made in recent decades to develop and use new materials and additives that are biodegradable and possess sustainable properties as well as a reduced carbon footprint by reducing greenhouse gases during their production [3,4,5,6].

In this context, until relatively recently, high performance polymers such as polyamides (PAs) were materials derived entirely from oil; however, new technologies and research have succeeded in obtaining them from monomers, both fully and partially renewable [7,8]. Monomers of biological origin include, for example, brazilic acid, sebacic acid, 1,4-diaminobutane (putrescine), and 1,5-diaminopentane (cadaverine) [9,10,11,12]. From these types of monomers, different kinds of polyamides can be obtained. It is always desirable to generate polymers with properties similar to those of petrochemical counterparts. This fact is vital for industry, because obtaining biobased polyamides (bio-PA) that behave similarly to PA6 and PA66 regarding stiffness, and similarly to PA12 regarding flexibility, is increasingly important for economic and ecological issues [13,14].

Currently, more than 6 million tons of polyamides are required annually, with growing demand [15]. In particular, polyamide 6 (PA6) and polyamide 66 (PA66) make up approximately 90% of the total polyamide use in the plastic industry [16]. However, as noted, the development of a “green” route for the production of biobased polyamides (bio-PA) has generated increasing interest due to the inevitable stoichiometric waste associated with the classic petrochemical production routes, which are commonly thought to cause global warming and other environmental problems [8,17]. For example, in some applications, PA6 can be replaced with a biobased variant called PA610, which possesses very similar properties, but it is more ductile as well as it exhibits high renewable content. Because of the fact that the dicarboxylic acid can readily be condensed with the petroleum-based 1.6-hexamethylenediamine (HMDA) obtained from butadiene, the material PA610, containing 60–63 natural content, may be obtained [18].

Within the polymer industry, polyamide 6 (PA6) is a highly relevant engineering polymer, which finds applicability in certain areas where high flame retardant and fire retardant properties are required [19]. The development of flame retardant additives that are attractive from an ecological point of view has become the focus of much effort. In particular, halogen-free flame retardants based on renewable sources have attracted great interest [20]. Many of these are phosphorus-based additives [21,22], such as aluminum hypophosphite (AlHP), with very good results for blends considering sustainable polymers such as polylactic acid [23] or polyvinyl alcohol [24], and others such as PA6 [25]. Other elements that are having large application as halogen-free retardants are ammonium polyphosphate (APP) or antimony trioxide, with great application at present in polyolefins [26,27]. Other materials like polyurethane foams, ammonium polyphosphate [28], or triphenyl phosphate (TPhP) [29,30] are often used as intumescent due to their low toxicity, those being halogen-free materials, and highly efficient. In other cases, flame-retardant additives containing elements other than phosphorus, such as expanded graphite [31], nanoclays, and nanosilicates [32,33] or graphite oxides [34], have been utilized. In this context, the search for this type of more natural additives has found elements such as halloysite nanotubes (HNTs) [35].

Blends containing halloysite nanotubes (HNTs) present great potential in the generation of natural fire retardancy polymeric materials distinguished by their high sustainability and low emission of toxic gases and fumes during combustion [36,37]. Normally, one of the main disadvantages of nanofillers is the low dispersion it presents in a polymer matrix, directly affecting both mechanical and flame-retardant properties [38,39]. However, because of their polar structure, HNTs can be efficiently dispersed in different polyamides [40,41]. Over the last two decades, nanocomposites based on inorganic clay minerals have attracted a lot of attention. In addition, due to their nanoscale structure, nanocomposites may exhibit significant improvements in aspects such as mechanical properties, reduced gas permeability, increased thermal stability, and improved flame retardancy compared to the properties of the polymer without these nanocomposites [23,24,25,26]. For this reason, the use of halloysite nanotubes is becoming more attractive. In this context, some authors have incorporated HNTs in polyamides of petrochemical origin such as PA 6 or PA66, providing promising results for blends between 5% and 40% of HNTs regarding mechanical properties and fire protection [42,43]. On the other hand, trying to get away from oil products, authors such as Sahnaoune et al. [44] are introducing these types of natural fillers into biobased polyamides such as Polyamide 11, taking into account as a disadvantage the fact that this type of polyamide is more expensive and has less applicability in today’s industry. For this reason, the aim of this project is to find a balance between sustainability and direct application in industry for Polyamide 610.

In previous works, the impact of the incorporation of HNTs into biobased PA610 at different levels on mechanical, thermal, and morphological properties has been deeply studied [45]. However, the main objective of the work reported here was the use of different levels of halloysite nanotubes (HNTs) to improve the flame-retardant properties of PA610. The objective was to determine how the incorporation of nanotubes into PA610 affects its fire retardancy and flame-retardant properties in high-performance injected parts. In order to determine the properties of the blends, several samples have been characterized using cone calorimetry, limiting oxygen index (LOI), and a calorimetric pump.

## 2. Materials and Methods 

### 2.1. Materials

Partially BioBased Polyamide 610 (PA610) was supplied by NaturePlast (Ifs, France), in the form of pellets. According to the manufacturer, this is a biobased medium-viscosity injection-grade homopolyamide with a density of 1.06 g/cm^3^ and a viscosity number (VN) of 160 cm^3^/g. This polyamide has 63% of biological content. As flame retardant, the halloysite nanotubes (HNTs) were supplied by Sigma Aldrich (Madrid, Spain) with CAS number 1332-58-7. This material had an average tube diameter of 50 nm and inner lumen diameter of 15 nm. Typical specific surface area of this halloysite was 65 m^2^/g.

### 2.2. Sample Preparation

Prior to processing, the biobased PA610 and Halloysite nanotubes were dried at 60 °C for 48 h in the dehumidifying dryer MDEO. Both components were mechanically pre-homogenized in a zipper bag. The materials were then fed into the main hopper of a co-rotating twin-screw extruder (Construcciones Mecánicas Dupra, S.L., Alicante, Spain). The screws featured 25 mm diameter with a length-to-diameter ratio (L/D) of 24. The extrusion process was carried out at 20 rpm, setting the temperature profile, from the hopper to the die, as follows: 215–225–235–245 °C. The different PA610/HNTs composites were extruded through a round die to produce strands and, subsequently, pelletized using an air-knife unit. In all cases, residence time was approximately 1 min. The four prepared compositions are shown in Table 1.

In the final step, the compounded pellets were shaped into square plates of 150 × 150 × 5 mm^3^ by injection molding in a Meteor 270/75 from Mateu and Solé (Barcelona, Spain). The temperature profile in the injection molding unit was 220 °C (hopper), 225 °C, 230 °C, and 235 °C (injection nozzle). A clamping force of 75 tons was applied while the cavity filling and cooling times were set to 2 and 20 s, respectively.

### 2.3. Material Characterization

#### 2.3.1. Cone Calorimeter Test (CCT)

The cone calorimeter model was 82121 (FIRE Ltd., Surrey, UK) and the tests were performed according to ISO 5660 standard procedures. The dimensions of the samples were 100 × 100 × 5 mm^3^. Each sample was wrapped in aluminum foil (0.0025 to 0.04 mm thick) and horizontally exposed to an external heat flux of 50 kW/m^2^, 25 mm conical distance, and 20 min test time.

#### 2.3.2. Limiting Oxygen Index (LOI) and UL94

LOI was carried out in an 82121 model (FIRE Ltd., UK) according to the standard oxygen index test stated in the UNE-EN ISO 4589-2 norm. Type I test pieces and the ignition procedure (A) related only to the upper surface were used. Prior to the test, the specimens were conditioned at 23 °C and 50% relative humidity for 24 h. The size of the samples used was 150 × 10 × 4 mm^3^. Three samples were studied using the LOI test. 

The UL-94 horizontal burn tests were carried out following the testing procedure UL 94:2006; EN 60695-11-10:1999/A1:2003 with a test specimen bar that was 150 mm long, 10 mm wide, and about 5 mm thick.

#### 2.3.3. Toxicity and Opacity Test

The smoke density chamber tests the opacity of the emitted fumes according to the EN ISO 5659-2 norm. This test allows to obtain the optical density using a simple chamber; simultaneously, the toxicity of the fumes was determined according to the UNE-EN 17084 norm. The test was carried out in a chamber model NBS Smoke Chamber (Concept Equipment Ltd., Arundel, UK), and a FTIR model MG2030 MKS Instruments, Inc. (San Diego, USA) to study the toxicity.

The dimensions of the samples were 75 × 75 × 5 mm^3^. The samples must be conditioned to a constant mass at a temperature of 23 °C and a relative humidity of 50% in accordance with ISO 291. Each sample was wrapped in aluminum foil (0.04 mm thick) and exposed to a radiation of 50 kW/m^2^, 25 mm conical distance, and 600 s test time. Regarding toxicity, a flow rate of 4 L/min was extracted for 30 s at minutes 4 and 8 of the test. Three replicates of each material were performed.

Regarding the calculation of the specific optical density (Ds), Equation (1) was used:(1)$$D_{s}\left(t\right)=\frac{V}{AL}\log_{10}\frac{100}{T\left(t\right)}\left[Adimensional\right]\;\;\;\;$$
where *Ds*(*t*) is the specific optical density; $\frac{V}{AL}$ is the ratio between the volume of the camera (*V*), the exposed area of the specimen (*A*), and the length of the light path (*L*). This ratio is equivalent to 132 and, finally, *T*(*t*) is the value of transmittance measured in %.

In relation to the toxicity, the concentration of toxic gas (CO_2_, CO, HF, HCl, HCN, NO_2_, SO_2_, HBr) and optical density of smoke were recorded following the previous norm (UNE-EN 17084). During the smoke toxicity test, the concentrations of eight toxic gases were used as quantifying terms for the conventional index of toxicity (CIT). CIT was calculated using Equation (2):(2)$$CIT_{G}=0.0805\overset{i=8}{\underset{i=1}{\sum }}\frac{c_{i}}{C_{i}}\;\;\;\;$$
where, *CIT_G_* was the conventional toxicity index for general products, *c_i_* was the concentration of the gas in the chamber, and *C_i_* was the reference concentration of the gas.

#### 2.3.4. Calorific Value

The equipment used for this test was a PARR 6200 calorimeter (Parr Instrument Company, Moline, IL, USA). The samples in pellet form were turned into fine powder by a grinder and liquid nitrogen was used in order to avoid thermal decomposition in the samples. The temperature of the distilled water was set at 26 °C and the closing pressure was set between 3.0 and 3.5 MPa with no air in the inside. The reagents used were distilled water, pressurized oxygen with purity greater than 99.5%, a standardized benzoic acid tablet, and a pure iron wire of 0.1 mm.

## 3. Results

### 3.1. Cone Calorimeter Test (CCT)

The calorimetric cone test (CCT) provides a great deal of information on the fire behaviour of the material under study when exposed to a variable radiation source [46]. The CCT is based on the principle of oxygen consumption, simulating the combustion of polymers in real fire situations, demonstrating great utility in research studies, and allowing the development of new materials with excellent fire-retardant properties [47,48]. Table 2 summarizes the main results obtained from this test in relation to thermal parameters and Table 3 summarizes the results of smoke parameters.

In relation to the ignition time of the samples (TTI), the incorporation of halloysite reduces the ignition start time by about 25 s in all the cases studied. Furthermore, the total duration of the inflammation does not present great variations, except for PA610/HTN10, which reaches values of 850s. Authors like Marney et al. [49] showed similar behaviour in PA6 mixtures with HNTs, where a reduction in the ignition time of the halloysite mixtures was observed. This factor may be due to the early release of the internal elements of the HNTs, which leads to the formation of small combustible molecules and results in an accelerated decomposition.

The flame retardant effect of aluminum phosphinate is in combination with zinc borate, borophosphate, and nanoclay in polyamide-6Mehmet. It is sometimes difficult to evaluate the performance of incorporating a flame-retardant additive because the results obtained are expressed in terms of time or released energy, thus generating problems of comparison. To overcome this issue, Vahabi et al. [50] have defined a dimensionless concept called the “Flame Retardancy Index” (*FRI*), which allows a very simple comparison between the pure polymer and its fire retardancy composite. This dimensionless concept is born from the following equation:(3)$$FRI=\frac{{\left[THR\cdot \frac{pHRR}{TTI}\right]}_{Neat\;Polymer}}{{\left[THR\cdot \frac{pHRR}{TTI}\right]}_{Composite}}\;\;\;\;\;$$

Thanks to the use of the *FRI*, a better comparison can be made between the different values obtained from the calorimetric cone test, obtaining a direct comparison between the pure polymer and its compounds. 

If the FRI values in Table 2 are analyzed, it can be seen directly whether the HNTs improve the fire-retardant characteristics. Initially, it is expected that by introducing the flame-retardant additive and dividing the term calculated for the neat polymer by that of its composite with HNTs, a dimensionless quantity greater than 1 is obtained. However, it can be seen how the incorporation of 10% of HNTs into the mixture reduces the value to 0.46, obtaining a very poor value. The incorporation of a higher amount of HNTs progressively improved the previous results. In particular, 30% of HNTs improve by more than 47% of the previous value, but both are below the unit. Following the guidelines of Vahabi’s work, any compound that has an FRI value below 1 is taken as the “Poor” level of performance in terms of fire retardancy. Following these general premises, HNTs seem to not provide a fire retardancy improvement when compounded with PA610. This factor may be related to the large fire-retardant capacity of PA, an engineering plastic with intrinsic fire retardancy properties, due to its base structure.

Some characteristic results of the calorimetric cone test are presented above.

#### 3.1.1. Heat Release Rate (HRR)

The HRR measured by cone calorimeter is a very important parameter as it expresses the intensity of fire [51]. In this section, Table 2 and Figure 1 show the results related to the heat released by the samples through the CCT. Referring to the maximum peak of heat released (pHRR), most of the samples exhibit a decrease in the heat released thanks to the presence of halloysite, with the exception of PA610/HTN10, which presents a higher peak than the base compound without additive, possibly due to a low concentration of the additive. This reduction of the peak indicates that little energy has been released to the system, verifying the possible application of HNTs as fire retardancy additives in certain applications. In relation to the time when the maximum pHRR value is produced, there seems to be no apparent relationship between this value and the amount of halloysite.

On the other hand, Figure 1 shows the evolution of the heat release rate as a function of time. A heat reduction can be observed for the first 10 min of the test, reaching maximum values around 300 s. The results show a slight increase in heat release in the PA610/HTN10 sample and a very slight reduction in the heat released in the PA610/HTN20 sample, both compared to the polymer without load. However, in the case of 30% halloysite, a significant reduction is achieved in terms of the maximum values obtained, going from 743 to about 580 kW/m^2^ and directly modifying the shape of the curve. This improvement in the heat released is produced by the carbonization of the halloysite, forming a layer that acts as a barrier that slows down the combustion of the polyamide, making it difficult for the sample to burn. In this context, authors like Marney et al. [42] showed similar results with the incorporation of HNTs in petrochemical polyamides. This happens due to the formation of a thin layer of carbon (or skin) on the surface, which breaks during the first stages of combustion, as shown by the small plateau at about 100, verifying how the modification in the shape of the curve is directly related to the formation of carbon in the external layers [52]. In addition, the HRR increases until it reaches a peak because of the increasing amount of combustible volatile compounds caused by the rise in total temperature of the substrate. The apparent temperature increases because the material properties are changing so that the unexposed surface reaches temperatures close to that of the exposed surface at the time of ignition. When the release of combustible volatiles has been exhausted, the combustion reaction ceases, and the peak is usually followed by a sharp linear decrease to zero.

The integration of the HRR curves allows to obtain the THR (total heat realised) values. The PA610 produced 128.1 MJ/m^2^; the introduction of HNTs slightly increased the heat to 164.3 for the PA610/20HNTs. The range of values obtained matches with the proposed by Doğan et al. [53] for a polyamide-6 and different flame-retardant additives.

#### 3.1.2. Effective Heat Combustion (EHC)

In the CCT, the effective heat of combustion (EHC) represents the heat released during combustion per unit mass. Figure 2 shows the graph of the results obtained as a function of time. A slight increase in the effective heat of combustion in the samples with HNTs can be seen, which is determined by the values of HRR and mass loss of the different materials during the test. Particularly, it can be seen how the PA610/10HNTs sample has provoked an increase of 20 MJ/kg compared to the sample of PA without any load. Qin et al. [54] showed similar results with the incorporation of montmorillonite to polyamide 66, where the average EHC value of PA66 increased after the addition of nano-loads, which were obtained from decomposition processes. This increase in EHC may be related to a low load concentration in the mixture, which generates an increase in the amount of energy released. This can be seen directly in samples where the amount of HNTs is higher, as it can be observed in 20% and 30% HNT samples, where the EHC is close to the PA610.

#### 3.1.3. CO and CO_2_ Production

Another important aspect to consider during a combustion produced by fire is the amount of CO and CO_2_ that is released from the burned products, mainly because a great release could cause anoxia conditions, making it difficult to evacuate the site. Table 3 shows the maximum production values for these gases. The analyzer of the equipment used is able to determine the concentration of CO and CO_2_ during the course of the test, making the analysis of the emission of these gases deeper.

Figure 3 shows the values of CO_2_ produced during the test in relation to the quantity released in kg per kg of material analysed and according to the release rate. Regarding the emitted quantity, a slight increase can be observed in the 20% and 30% HNT samples compared to the base material, while in the case of the 10% HNT sample, the emission of CO_2_ doubles the quantity emitted by the base polyamide. On the other hand, referring to the release rate, although it seems that the amount of total CO_2_ emitted is greater as the additive load is increased, as observed with the 10% and 20% HNT samples, a significant reduction in the maximum peak for the CO_2_ rate in the case of the 30% HNT sample can be observed. This fact could mean that the multi-layered porous nature of the HNT structure may prevent those evolved gases from entering the combustion zone of the burned sample. In addition, the heat feedback from the flame zone to ensure a faster polymer decomposition may also be restricted by the insulating nature of the carbon structure in the HNTs [55].

In relation to the values obtained for CO, Figure 4 shows how the values obtained both in total production and ratio are relatively low for all samples. However, from a global point of view, it can be seen how the incorporation of HNTs into the PA610 matrix means a slight decrease in the amount of CO generated during combustion. However, these values are not as representative as those obtained for the CO_2_ produced. This factor indicates that the compounds are burned reasonably efficiently (since carbon monoxide can be measured due to an incomplete combustion) [42]. In this sense, Gilman et al. [56] studied silicate nanocomposites in PA6 and suggested that if EHC, smoke extinguishing area (SEA), and CO yields did not change, flame inhibition of the condensed phase was involved, and this was accompanied by a decrease in PHRR and mass loss ratio (MLR) as well as a change in carbon yield.

#### 3.1.4. Rate of Smoke Production (RSP)

The smoke performance of a fire retardancy material is a vital parameter in terms of fire safety. In the case studied, the significant increase in the amount of smoke produced is related to HNTs incorporation into the blends. This increase can be linked to the formation of a carbonized layer of material during combustion produced by halloysite itself. It is this outer layer on the surface of the test piece that could cause the combustion to release a greater amount of smoke. It should be noted that an increase in the amount of carbonized residue is observed at the end of the test in materials with halloysite. The appearance of carbonized combustion products generates an emission of darker fumes, which causes an increase in the RSP as shown in Figure 5. Similar results were reported by Levchik et al. [57] where they showed that the amount of smoke produced by the incorporation of HNTs was almost double than that of the original polymer because HNTs appear to aid the process of carbon formation in the composite material by acting as a “glue” between the HNTs, thus ensuring the formation of a consistent and strong porous carbon layer over the polymer.

The increase in the RSP value during the test is strongly related to the smoke formation as determined by the smoke extinguishing area (SEA) (Figure 6) during the combustion of HNT-containing compounds; however, it does not seem to depend on the amount of HNT. The presence of HNTs seems to accelerate the rate of PA610 smoke production during the first stages of the combustion process, being that this effect is especially marked in the 10% HNT sample. Similar results were reported by Marney et al. [42]. 

#### 3.1.5. MASS 

Regarding the amount of final mass obtained after the test, Figure 7 shows the results obtained in terms of mass loss ratio and percentage of residual mass. In particular, Figure 7a shows an increase in residual mass with the incorporation of HNTs. This increase is closely related to the amount of carbonized mass remaining as waste at the end of the test, which is largely made up of carbonized halloysite. The creation of this layer of carbonized material has been reported by various authors even with the use of other flame retardants such as APP or TiO_2_ [58]. An intumescent carbon layer may appear on the surface of materials during combustion, creating a physical protective barrier able to contain heat and mass transfers. The carbon layer limits the diffusion of oxygen to the underlying part of the material or insulates it from heat and combustible gases; it also further delays the pyrolysis of the material. The mass loss decrease was attributed to the formation of carbon and the morphological structure at the surface of the materials [59,60]. Similarly, the result of each test specimen before and after exposure to cone radiation can be seen in Figure 8. In the case of PA610, total disappearance of the material can be seen, while in the rest of the tests, a layer of carbonized material appears, formed mainly by halloysite remains. These remains may be evidence of the increase in smoke generation previously observed. 

In this sense, the incorporation of HNTs in relation to the residual mass is closely related to the values obtained in TGA degradation behavior in the previous work [45], focused on mechanical, thermal, morphological, and thermomechanical properties. The previous TGA results on the PA610/HNT system indicated a slight increase in the onset degradation temperature from 417.4 °C up to 419.6 °C, as well as the maximum degradation rate temperature (from 461.5 °C up to 466.6 °C). The residual weight is, as expected, close to 10, 20, and 30 wt.%, as HNTs do not decompose below 700 °C. In the previous work, scanning electron microscopy images (FESEM) also revealed good particle dispersion (especially for 10 wt.% and 20 wt.% HNTs). It can be seen how the residual mass of the compositions are almost coincident with the nominal HNTs content. 

On the other hand, Figure 7b shows the mass loss ratio (MLR), which is a relevant factor in terms of selecting a fire retardancy additive, as it provides valuable information about the behavior against fire of the analyzed materials. The results obtained in the MLR are closely related to those observed in the smoke extinction area (SEA) in Figure 6, where the samples with halloysite present lower mass loss than the PA610 samples. Some authors have shown how the HNTs reduce the fire hazard of the butadiene-acrylonitrile rubber (NBR) vulcanizates. They clearly extend the time to ignition (TTI), and positively influence the average mass loss rate (MLR) parameter. Because of the synergetic relation between flame retardants and halloysite nanotubes, the parameters connected with the amount of heat released during NBR composites’ combustion are reduced [36].

#### 3.1.6. The Maximum Average Heat Rate of Emission (MARHE)

The maximum average heat emission index (MARHE) is a parameter used in the EN 45545-2:2013+A1:2015 standard to classify the materials to be tested for railway applications. This value is obtained by dividing the maximum heat emission value recorded (in kW) by the area of the test specimen (0.01 m^2^). The maximum heat emission value per unit area (kW/m^2^) alongside other results such as opacity and smoke toxicity are used to classify the tested material according to the applicable risk stated in the standard norm stated above. In this context, Table 4 shows MARHE’s results obtained in the tests carried out.

The values obtained in all cases were very high. The PA610 shows values above 335 kW/m^2^ and the incorporation of HNTs increases this value up to 409 kW/m^2^ for a concentration of 10% HNTs. In the European context, the MARHE value must be inferior to 90 kW/m^2^ for these materials to be used in railway applications; the green composites studied here exhibit higher MARHE values (330–400). Additional flame-retardant additives or protective coatings for these biocomposites should be able to reduce their MARHE value, but for the time being, these green composites are not suitable for European rail applications due to the limitation exposed [61].

### 3.2. Limiting Oxygen Index (LOI) and UL94 Results

This test is defined as the minimum percentage of oxygen needed in a mixture in order to maintain the combustion of the sample after ignition. LOI tests are widely used to evaluate fire-retardant properties of materials, especially for the screening of fire-retardant polymer formulations. In this context, Figure 9 shows the LOI values obtained for PA610 with different concentrations of HNTs in their structure. 

The mean of the results obtained for the PA610 without HNTs stands at 27.2%. This value is relatively high for this type of polymer. Other authors have reported values close to 20–25% for PA6 [42,53], verifying the good application that this type of biopolyamide can have from the point of view of flammability. The inclusion of HNTs means a slight decrease in the LOI values, standing at 26% for a 30% load of HNTs. Authors like Li et al. [62] showed how the incorporation of 2% HNTs into PA6 slightly improved the LOI values of this polymer, but always remained in values below 25%. In general, from the experimental results obtained for LOI, a decrease in the oxygen limit value could be observed. This means that the increase of halloysite in the polyamide makes the combustion of the material easier in low oxygen concentration conditions. In spite of this fact, variations in numerical results do not suppose a considerable change leaving aside the facility for ignition of the compound (differences smaller than a 2% of concentration in volume of O_2_). Similar results have been reported by Sol et al. [63], where the increase of HNTs in the mixtures supposed a slight decrease in the LOI values, but always stayed at values superior to 24% of LOI. 

The incorporation of HNTs in large quantities does not improve flammability for PA610 due to its good initial results. Authors like Vahabi et al. [64] have reported that the best results of HNTs as flame retardants are found in percentages close to 10%, verifying the results obtained in this experiment relating to LOI values.

The UL-94 test shows that all the samples comply remarkably with the test, providing in all cases a V-0 classification. This factor verifies that the incorporation of HNT does not impair the original properties of PA610, showing promising results.

### 3.3. Smoke Density and Toxicity Analysis

Among the parameters that can be obtained by means of the CCT, the specific optical density (Ds) is a measurement of the degree of opacity of smoke, taken as the negative decimal logarithm of the relative light transmission. Figure 10 shows the optical density evolution along time. Except for the 20% HNTs mixture, the incorporation of nanotubes generates a slight decrease in the maximum values obtained for optical density as the halloysite load in the polyamide matrix increases. In addition, an increase in the time needed to generate smoke can be seen in the samples with HNTs. This type of behavior is very important in applications where properties against fire are needed, verifying the applicability of these types of natural loads in different fields.

In addition, the representation of density evolution as a function of time is shown according to the UNE-EN ISO 5659 standard. It is also important to point out the specific optical density values after 10 min of testing, as well as the maximum value (Ds_max_). In this context, Table 5 shows values obtained by Equation (1). These values are closely related to those that can be seen in Figure 10.

The results obtained here are very similar to those obtained by Zhang et al. [65]. These authors reported that the addition of 5% in weight of HNT in a thermosetting resin managed to reduce the density of the smoke emitted by the mixtures. This result means that the incorporation of this additive generates a synergetic effect on the suppression of smoke from the resin, PA610 being the case studied. The reduction in optical density may be related to good dispersion of HNTs in the PA610 matrix. As already verified by mechanical properties, reducing agglomerates makes the HNTs act more adequately.

The smoke toxicity test procedure for railway industry applications follows EN 45545-2/ISO 5659-2 standards [66] performed at 50 kW/m^2^. Related to toxicity, one of the main causes of death in cases where fire is involved is toxic gases generation [48]. Table 6 shows the results obtained in terms of toxicity of the analysed fumes, such as CO_2_, CO, HCl, HF, HCN, NO_2_, SO_2_, and NO. It can be seen how the incorporation of HNTs produces, in most cases, a clear reduction in the emission of certain gases, such as CO_2_ and NO_2_. In particular, it can be seen how the 30% HNTs mixture reduces this value from 437.6 kg/kg to 151.2 kg/kg in the case of CO_2_. Those are quite promising values referring to this type of material. Attia et al. [67] showed similar gas reduction results with the incorporation of different HNT loads in an ABS matrix. In addition, certain charges present in silicates, such as iron oxides, can participate in the flame-retardant mechanism by trapping radicals during polymer degradation, thus improving thermal stability and flammability properties for these nanocomposites [68,69].

In order to make a deeper analysis of the final toxicity of the samples, the CIT_G_ was obtained. This value is a dimensionless index that provides information about the overall toxicity of all the combustion gases analyzed.

Table 7 shows the results obtained from the CIT_G_ index. It can be seen that most of the samples with halloysite present a notable reduction in emitted gases concentration compared to pure PA610. However, the 10% HNTs sample shows a slight increase in the final concentration of emitted gases, which results in a higher index in comparison to the base material (PA).

### 3.4. Calorific Value

The calorific value test is obtained by means of a calorimetric pump. It is used to establish the combustion power in MJ/kg of the material to be characterized, and to analyze the existing difference made by the incorporation of fire retardancy additives. To carry out this test, the material is first introduced in a crucible together with an ignition wire, inside an airtight container with oxygen under pressure. It is then filled with pressurized oxygen gas. Next, the determination of combustion energy is carried out, so it is introduced inside a container with water and two electrodes are connected to each side of the previously introduced ignition wire. Finally, by means of agitation made by the equipment, the temperature of water is maintained homogeneous so that the temperature probe determines the increase in centigrade degrees of water produced by the material combustion.

Table 8 shows combustion energy and temperature difference values for the characterized samples. It can be seen how the incorporation of HNTs supposes a clear reduction of the combustion heat, verifying the application of these types of natural loads as fire retardancy additives. The combustion energy value for PA610 is 33.2 MJ/kg, while the samples with HNTs as additives, particularly the mixture with 30% HNTs, reduce the combustion value to 23.9 MJ/kg. This reduction is a clear advantage of this type of nanocomposites, since they have direct impact on flammability. During combustion, a halloysite-rich barrier is formed, delaying mass loss/transfer (i.e., less fuel available in the flame zone). Additionally, the refractory nature of the HNT limits the heat conduction and results in the reduction of available energy, as proposed by Marney or Smith [42,70]. There have been proposed several fire retardancy mechanisms that HNTs can exert. One of the proposed mechanisms is the typical formation of a char layer that prevents direct contact of the polymer with oxygen and slows down the escape of volatile products. Another mechanism considers that iron oxides contained in HNTs can trap free radicals during the decomposition, thus allowing to delay the burning process as observed in other iron-containing nanoparticle systems for improved flammability [71]. Nevertheless, the amount of iron oxide in HNTs is relatively low (0.29 wt.%), and there would be not enough iron oxide to play main role in fire retardancy with HNTs [72]. Du et al. [68] reported that the particular nanotube structure of HNTs allows entrapment of the decomposition products in the lumen, with a positive effect on delaying the mass transport and thus, leading to increased thermal stability. This phenomenon that allows to improve the fire retardancy properties has been represented in Figure 11. Authors like Hajibeygi et al. [73] obtained similar energy values for polyamide, besides corroborating the reduction of heat release thanks to the incorporation of additives such as zinc oxide (ZnO) nanoparticles. On the other hand, regarding the incorporation of natural nanocomposites as fire retardancy additives, Majka et al. [74] showed very similar heat release results for polyamide 6 with montmorillonite (MMT), justifying the incorporation of natural additives as fire retardancy elements.

This improvement in heat release may be closely related to the good dispersion obtained by HNTs in PA610. This factor can be seen directly in the mechanical properties section, and in particular, in the morphology of the samples from the previous work [45].

## 4. Conclusions

This work shows that the incorporation of nano-loads with fire-retardant properties, such as HNTs, can be effectively used as new reinforcement elements in partially biobased PA610 parts prepared by conventional industrial processes for thermoplastic materials, such as injection molding. 

In relation to cone calorimetric values, the incorporation of 30% HNTs achieved a significant reduction in terms of the maximum values obtained of HRR, changing from 743 kW/m^2^ to about 580 kW/m^2^ and directly modifying the shape of the curve. This improvement in the heat released is produced by the entrapment of the volatile decomposition products into the HNTs’ lumen, which slows down the combustion of the polyamide, making it difficult for the sample to burn. However, in relation to ignition time of the samples (TTI), the incorporation of HNTs reduces the ignition start time about 20 s. This factor causes the FIR value of the compounds with HNTs to be below 1, showing poor behavior in terms of fire-retardant characteristics. In addition, the incorporation of these natural nano-loads favors the delay of the time needed to provoke an increase in generated smoke. From the point of view of gas emission and average toxicity, the CITg index, the incorporation of 20 and 30% HNTs, generates a clear reduction in emitted gases concentration, close to 50%, compared to neat PA610.

On the other hand, considering released energy during combustion, a clear reduction can be seen in all the cases where HNTs have been added as nano-loads. The incorporation of 30% of HNTs generates a reduction of 9 MJ/kg of energy released in relation to pure PA610. The good dispersion analysed in the previous work between the HNTs and the PA610 matrix could be the cause of the good results obtained in energy release. Because it prevents HNTs agglomerates from being generated and avoids the loss of fire properties.

The presence of HNTs in the PA610 matrix means a notable reduction in the number of toxic gases emitted. Especially in the case of the 30% HNTs mixture, which reduces the quantity of CO_2_ emitted from 437.6 kg/kg to 151.2 kg/kg. The results obtained indicate that it is possible to obtain compounds with high renewable content such as PA610, and natural inorganic filler with nanotube structure, that is, HNTs, which exhibit acceptable fire-retardant properties, at the same time that it is possible to obtain a highly balanced material from the mechanical, thermal, and flame-retardant point of view. This improvement in properties against fire is very relevant in many applications where fire safety is crucial.

## Figures and Tables

**Figure 1 polymers-12-03050-f001:**
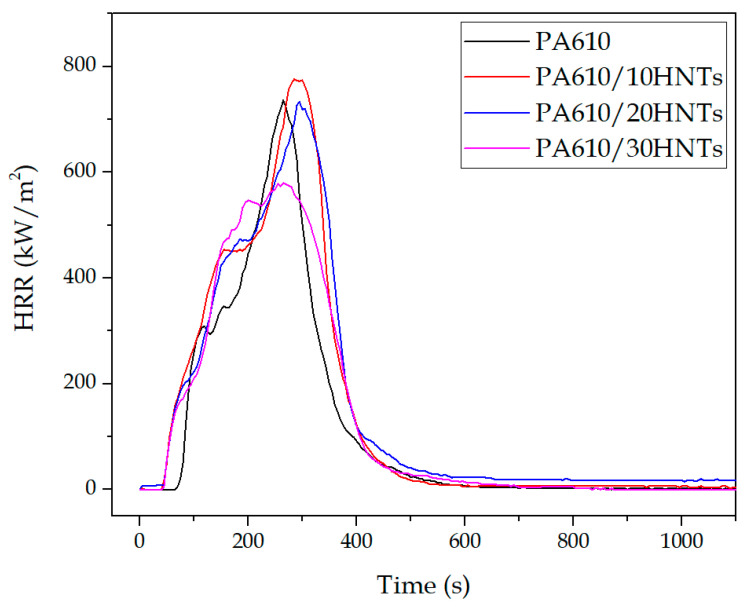
Heat release rate as a function of time.

**Figure 2 polymers-12-03050-f002:**
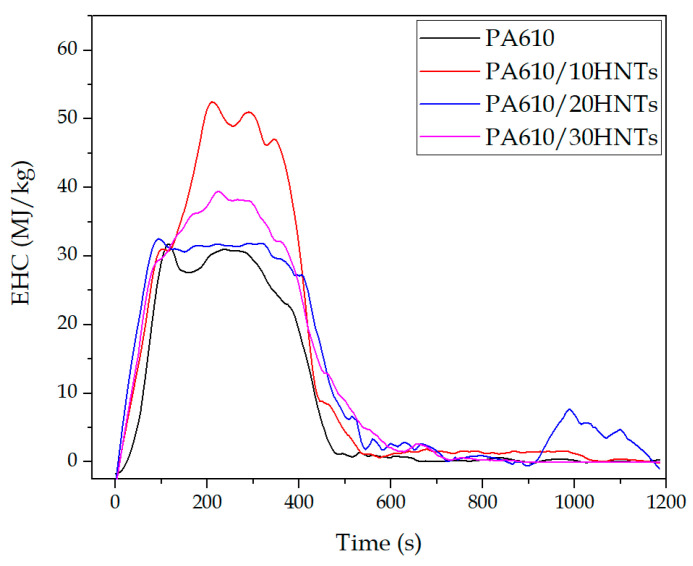
Effective heat of combustion as a function of time.

**Figure 3 polymers-12-03050-f003:**
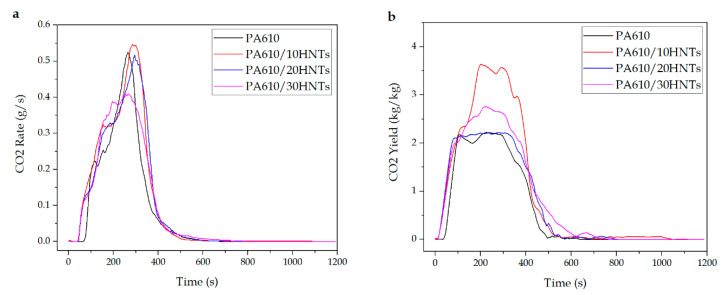
CO_2_ production during the CCT test: (**a**) CO_2_ rate and (**b**) CO_2_ yield.

**Figure 4 polymers-12-03050-f004:**
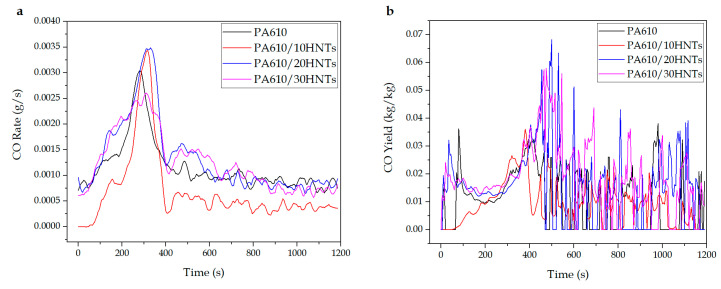
CO production during the CCT test: (**a**) CO rate and (**b**) CO yield.

**Figure 5 polymers-12-03050-f005:**
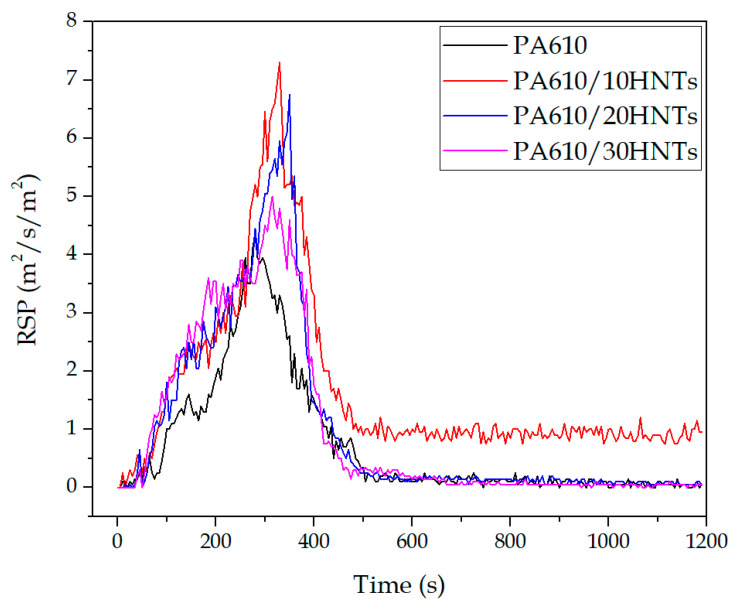
Smoke production rate of samples with HNTs.

**Figure 6 polymers-12-03050-f006:**
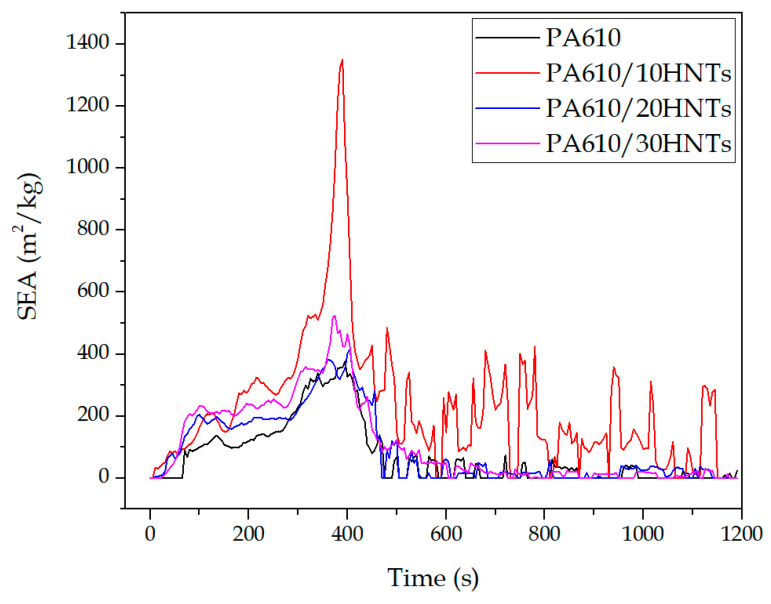
Specific extinction area for PA610 samples loaded with HNTs.

**Figure 7 polymers-12-03050-f007:**
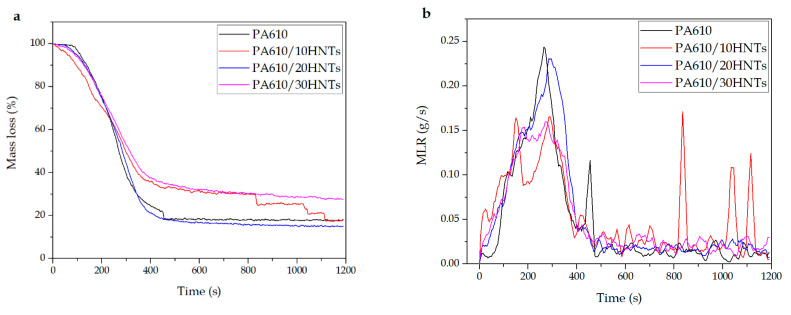
(**a**) Percentage of residual mass during the testing of PA610 samples with HNTs; (**b**) Loss of mass ratio of PA610 samples with HNTs.

**Figure 8 polymers-12-03050-f008:**
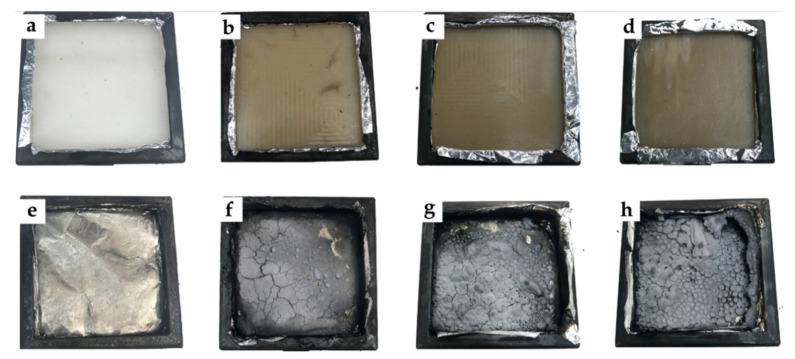
Visual difference between samples before and after the CCT test: (**a**) PA1010 before, (**b**) PA1010 /10HNTs before, (**c**) PA1010/20HNTs before, (**d**) PA1010/30HNTs before and (**e**) PA1010 after, (**f**) PA1010/10HNTs after, (**g**) PA1010/20HNTs after, (**h**) PA1010/30HNTs after.

**Figure 9 polymers-12-03050-f009:**
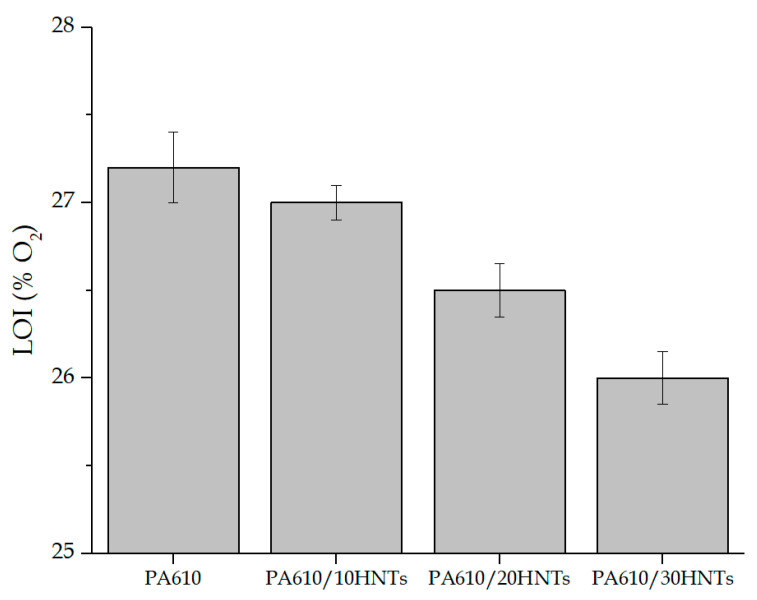
Graphic representation of the limiting oxygen index (LOI) values of each sample.

**Figure 10 polymers-12-03050-f010:**
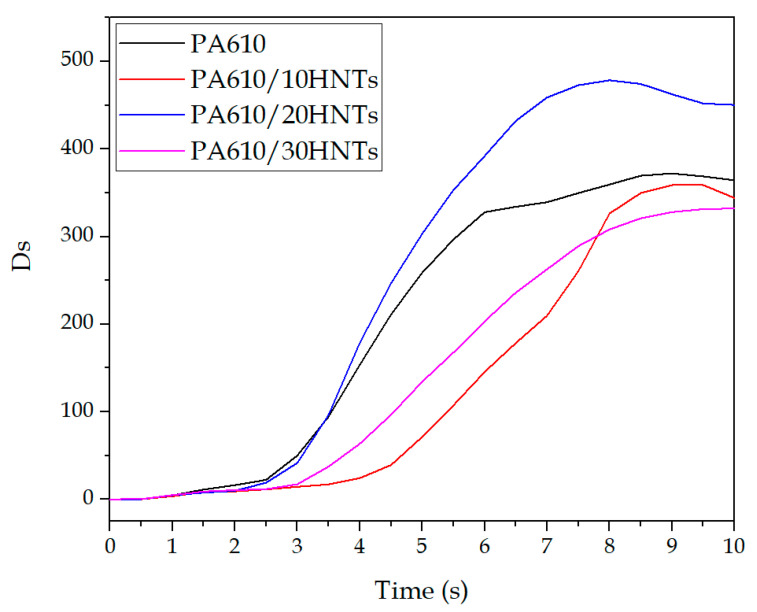
Evolution of optical density as a function of time.

**Figure 11 polymers-12-03050-f011:**
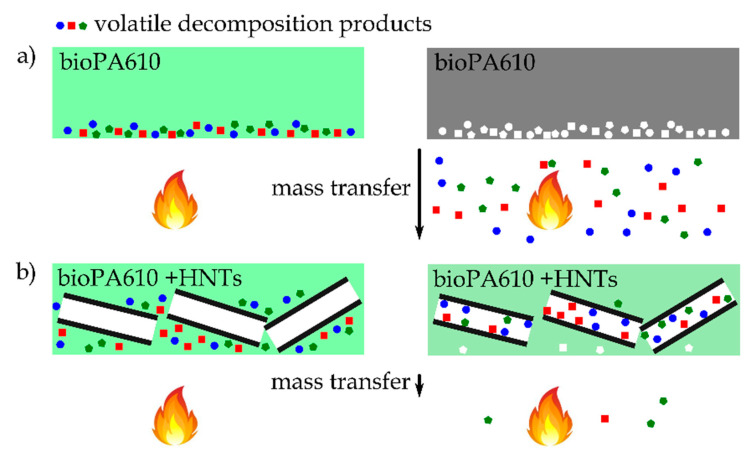
Scheme of the fire retardancy enhancement by HNTs by the entrapment of volatile decomposition products and delaying the mass transfer in (**a**) bioPA610 and (**b**) bioPA610+HNTs.

**Table 1 polymers-12-03050-t001:** Summary of compositions according to the weight content (wt.%) of polyamide 610 (PA610) and halloysite nanotubes (HNTs).

Code	PA610 (wt.%)	HNTs (wt.%)
PA610	100	0
PA610/10HNTs	90	10
PA610/20HNTs	80	20
PA610/30HNTs	70	30

**Table 2 polymers-12-03050-t002:** Summary of thermal parameters obtained with the calorimetric cone test (CCT) on the PA610 and HNTs samples.

Code	TTI(s)	t_sos._ _inflammability_ (s)	pHRR (kW/m^2^)	tpHRR (s)	EHC (MJ/kg)	THR (MJ/m^2^)	FRI
PA610	73.5 ± 0.5	621 ± 3	743 ± 4	272 ± 3	31.7 ± 1.9	128.1 ± 10.2	1 ± 0.1
PA610/10HNTs	45.5 ± 0.3	853 ± 4	800 ± 10	290 ± 4	52.2 ± 2.4	160.8 ± 8.3	0.46 ± 0.2
PA610/20HNTs	47.0 ± 0.4	694 ± 4	738 ± 10	300 ± 2	32.3 ± 1.6	164.3 ± 7.4	0.50 ± 0.3
PA610/30HNTs	45.0 ± 0.2	695 ± 3	581 ± 8	268 ± 2	39.3 ± 1.5	147.0 ± 9.9	0.68 ± 0.2

**Table 3 polymers-12-03050-t003:** Summary of smoke parameters obtained with the CCT on the PA610 and HNT samples.

Code	SEA (m^2^/kg)	CO_2_ Yield_max_ (kg/kg)	CO Yield_max_ (kg/kg)	Total Smoke (m^2^/m^2^)
PA610	360 ± 15	2.2 ± 0.5	0.037 ± 0.003	915.5 ± 16.5
PA610/10HNTs	1344 ± 32	3.6 ± 0.3	0.035 ± 0.002	1993.1 ± 25.1
PA610/20HNTs	403 ± 12	2.2 ± 0.4	0.067 ± 0.004	1245.4 ± 23.6
PA610/30HNTs	517 ± 26	2.7 ± 0.2	0.056 ± 0.002	1190.1 ± 19.8

**Table 4 polymers-12-03050-t004:** Summary of results of maximum average heat emission index (MARHE).

Code	MARHE (kW/m^2^)
PA610	337.8 ± 5.2
PA610/10HNTs	409.4 ± 8.1
PA610/20HNTs	396.9 ± 7.5
PA610/30HNTs	363.4 ± 9.8

**Table 5 polymers-12-03050-t005:** Summary of results of the maximum specific optical density values (Ds_max_) and at 10 min of testing (Ds_10_).

Code	Ds_10_	Ds_max_
PA610	364.3 ± 4.2	372.2 ± 5.9
PA610/10HNTs	344.2 ± 5.5	358.8 ± 4.8
PA610/20HNTs	450.5 ± 6.1	478.5 ± 5.4
PA610/30HNTs	332.1 ± 4.0	332.1 ± 4.5

**Table 6 polymers-12-03050-t006:** Volumetric fraction of the combustion gases.

	Volumetric Fraction (µL/L)							
	PA		PA/10HNTs		PA/20HNTs		PA/30HNTs	
	4 min	8 min	4 min	8 min	4 min	8 min	4 min	8 min
CO_2_	169.51	437.59	253.99	479.68	114.16	102.94	112.31	151.19
CO	0.18	0.10	2.83	2.97	0.17	0.01	0.36	0.23
HCl	0.11	0.15	0.12	0.33	0.02	0.02	0.27	0.20
HF	0.05	0.04	0.16	0.17	0.05	0.07	0.03	0.06
HCN	0.16	0.03	0.81	0.95	0.38	0.46	0.12	0.06
NO_2_	0.42	0.58	0.06	0.81	0.31	0.16	0.29	0.29
SO_2_	33.15	29.38	58.73	57.71	22.85	22.51	25.63	26.03
NO	0.36	0.65	0.23	1.22	0.15	0.13	0.18	0.25

**Table 7 polymers-12-03050-t007:** Summary of the results of the conventional index of toxicity (CIT_G_) index.

Code	CIT_G_	
	4 min	8 min
PA610	0.125 ± 0.005	0.111 ± 0.005
PA610/10HNTs	0.221 ± 0.005	0.220 ± 0.005
PA610/20HNTs	0.087 ± 0.005	0.085 ± 0.005
PA610/30HNTs	0.096 ± 0.005	0.098 ± 0.005

**Table 8 polymers-12-03050-t008:** Summary of calorific values results obtained in the PA610/HNT samples.

Code	Heat Release (MJ/kg)	∆T (°C)
PA610	33.2 ± 0.2	1.7 ± 0.1
PA610/10HNTs	30.2 ± 0.1	1.5 ± 0.1
PA610/20HNTs	27.3 ± 0.1	1.4 ± 0.1
PA610/30HNTs	23.9 ± 0.2	1.2 ± 0.1

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
