# Peer review of "The Effect of Halloysite Nanotubes on the Fire Retardancy Properties of Partially Biobased Polyamide 610"

_polymers, 2020, doi:10.3390/polym12123050_

Round 1
Reviewer 1 Report
There are still some minor points to be calified.
-Usually, we use "TTI" for the abbreviaition of time to ignition. please replace "ting" by TTI
in FRI formula, there is a mistake. Please check again: it should be "THR" instead of "EHC", the unit of THR is "MJ/m²"...
With this modification I think you will have FRI more than 1.
Author Response
Reviewer 1
There are still some minor points to be calified.
-Usually, we use "TTI" for the abbreviaition of time to ignition. please replace "ting" by TTI
ANSWER
Thank you for the comment. The manuscript has been revised and ting has been changed by TTI.
in FRI formula, there is a mistake. Please check again: it should be "THR" instead of "EHC", the unit of THR is "MJ/m²"...
With this modification I think you will have FRI more than 1.
ANSWER
Thank you for the recommendation. The revised manuscript has been modified, and the FRI equation has been checked. For this propose, THR values have been calculated. Despite the modifications proposed, FRI still been under the unit.
Table 2. Summary of thermal parameters obtained with the CCT on the PA610 and HNTs samples.
Code |
TTI(s) |
tsos. inflamability(s) |
pHRR (kW/m2) |
tpHRR (s) |
EHC (MJ/kg) |
THR (MJ/m2) |
FRI |
PA610 |
73.5±0.5 |
621±3 |
743±4 |
272±3 |
31.7±1.9 |
128.1±10.2 |
1±0.1 |
PA610/10HNTs |
45.5±0.3 |
853±4 |
800±10 |
290±4 |
52.2±2.4 |
160.8±8.3 |
0.46±0.2 |
PA610/20HNTs |
47.0±0.4 |
694±4 |
738±10 |
300±2 |
32.3±1.6 |
164.3±7.4 |
0.50±0.3 |
PA610/30HNTs |
45.0±0.2 |
695±3 |
581±8 |
268±2 |
39.3±1.5 |
147.0±9.9 |
0.68±0.2 |
Reviewer 2 Report
The Authors have revised the manuscript according to most of the Reviewers' comments and suggetions. Now the manuscript seems to be suitable for publication in Polymers
Author Response
Thank you for the revision. We really appreciate your comment.
This manuscript is a resubmission of an earlier submission. The following is a list of the peer review reports and author responses from that submission.
Round 1
Reviewer 1 Report
HNT is a nano-filler with high cost. However, the loading of HNT is 10-30 wt% in this work. Did the authors consider the cost of these composites? The improvement of flame retardant properties is very limited. For example, the LOI value decreased with the increasing concentration of HNT. The addition of HNT didn’t efficiently reduce the release of CO and HCN. Based on the high loading and limited flame retardant effect, it makes no sense to use HNT as flame retardant fillers. In addition, there are no characterizations of dispersion and TGA. The data used in this work cannot support the conclusions. As a result, I think this manuscript is not suitable to be published in POLYMERS.
Reviewer 2 Report
The topic is interesting. My comments are below:
- “LOW environmental impact” cannot be the main objective. Nowadays, There are several unsolved question regarding the impact of nanofillers…
- “fireproof” is not a correct term, it should be replaced by “flame or fire retardancy”
- In this sentence “were obtained by fusion extrusion in a twin-screw co-rotating extruder.”, please delete “fusion extrusion”
- Some quantitive data, especially from cone cal., should be added to the abstract
- High-performance fire-retardant polyamide materials, in Novel Fire Retardant Polymers and Composite Materials, this paper contains some information in regard with biobased PA. and also “Halloysite nanotubes (HNTs)/polymer nanocomposites: thermal degradation and flame retardancy” for HNT in “Clay Nanoparticles Properties and Applications”
- In Table 2, FRI Flame Retardancy Index (Flame Retardancy Index for Thermoplastic Composites, Polymers 2019) can be calculated and add to these table. Authors should discuss about it in the main text. The FRI gives the possibility to compare easily all samples. (https://www.mdpi.com/2073-4360/11/3/407)
- The mechanism of action behind flame retardancy can be explored better. A paragraph before the conclusion with a scheme will be informative and let readers to have a global view on mechanism.
Reviewer 3 Report
The paper from Marset et al. reports on the preparation and characterization of PA610 compounds containing 10, 20, and 30 wt. % of halloysite nanotubes, aiming at investigating the effect of the inorganic filler on the flame retardant properties of the partially biobased polyamide. The manuscript is quite well written and the conclusions are quite well supported by the experimental data.
Minor concerns:
- Table 2: too many decimal digits for TTi and the other thermal parameters.
- Table 4: K is not in capital letter
- It would be reasonable to make some SEM-EDX analyses on the residues after cone calorimetry tests and make some comments on the morphology of the burnt materials. This would help to better understand the flame retardant mechanism exerted by halloysite nanotubes